# Synthesis of Hydrophobic Nanosized Silicon Dioxide with a Spherical Particle Shape and Its Application in Fire-Extinguishing Powder Compositions Based on Struvite

**DOI:** 10.3390/nano13071186

**Published:** 2023-03-27

**Authors:** Igor V. Valtsifer, Yan Huo, Valery V. Zamashchikov, Artem Sh. Shamsutdinov, Natalia B. Kondrashova, Anastasia V. Sivtseva, Anna V. Pyankova, Viktor A. Valtsifer

**Affiliations:** 1Institute of Technical Chemistry, Ural Branch, Russian Academy of Sciences—Perm Federal Research Center, Russian Academy of Sciences, 614013 Perm, Russia; 2College of Aerospace and Civil Engineering, Harbin Engineering University, Harbin 150001, China; 3Institute of Chemical Kinetics and Combustion, Siberian Branch, Russian Academy of Sciences, 630090 Novosibirsk, Russia

**Keywords:** nanosized silicon dioxide, spherical monodisperse particles, superhydrophobicity, fire-extinguishing powder composition, struvite, powder flowability

## Abstract

Textural and morphological features of hydrophobic silicon dioxide, obtained by the hydrolysis of tetraethoxysilane in an ammonia medium followed by modification of a spherical SiO_2_ particles surface with a hydrophobic polymethylhydrosiloxane, were studied in this work. The size of silicon dioxide particles was controlled during preparation based on the Stöber process by variation of the amount of water (mol) in relation to other components. The ratio of components, synthesis time and amount of the hydrophobizing agent were determined to obtain superhydrophobic monodisperse silicon dioxide with a spherical particle size of 50–400 nm and a contact angle of more than 150°. In the case of the struvite example, it was demonstrated that the application of spherical- shaped hydrophobic silicon dioxide particles in powder compounds significantly improves the flowability of crystalline hydrates. The functional additive based on the developed silicon dioxide particles makes it possible to implement the use of crystalline hydrates in fire-extinguishing powders, preventing agglomeration and caking processes. The high fire-extinguishing efficiency of the powder composition based on struvite and the developed functional additive has been proven by using thermal analysis methods (TGA/DSC).

## 1. Introduction

Silicon dioxide is widely known for its application in various fields of industry, medicine, and scientific research connected with the role of silica in nature, its high biocompatibility and its chemical stability. Free silanol groups (Si-OH) on the surface of silica can interact with various inorganic and organic compounds, imparting desired functional properties to materials [1,2,3,4,5,6,7,8]. The number of publications in the scientific press devoted to silica surface modification mechanisms and the use of modified silica materials in various fields of chemistry and medicine are growing every year.

A specific fundamental and practical interest of researchers is dedicated to the preparation of superhydrophobic surfaces with a water contact angle of more than 150° [1]. Such materials, due to their self-cleaning ability, can be used for protection from various contaminants; for sorption processes, biologically active substances including oleophilic compounds; for solving environmental problems, such as the separation of water-oil emulsions, etc.

In the investigation [9], various types of hydrophobic (hydrophilic) silica sols were prepared and modified with polysiloxane and dexamethyldisilazane using tetraethoxysilane as a starting material. The surface-energy inhomogeneity of silicon dioxide (Aerosil) was studied before and after its surface modification by grafting various amounts of hexadecyltrichlorosilane [10]. Silica surface wettability after modification with n-octadecyltrichlorosilane was investigated in [11]; the authors also estimated the effect of temperature on the silanization rate. The study [12] describes a method for the preparation of superhydrophobic mesoporous structures with an application of trifunctional alkoxysilane precursors (methyltrimethoxysilane and methyltriethoxysilane) containing a hydrophobic methyl group. Currently, a large number of works are dedicated to the utilization of modified silica materials as functional additives for powder compositions to improve the flowability of cohesive powders and impart hydrophobic properties to them [13,14,15,16].

The negative effect of air humidity can be controlled by creating a dry coating of hydrophobic silica particles [15,16]. On the one hand, a coating of nanosized silicon dioxide particles increases the distance and diminishes the contact area between the particles of powder composition, reducing the surface energy and electrostatic charge [13]. On the other hand, the coating particles are able to absorb water vapor and prevent the formation of capillary bridges between the components [14]. Thus, it is possible to achieve a free-flowing state of powder composition.

In our previous investigations [17,18], hydrophobized silicon dioxide was considered a functional additive to improve the performance properties of fire-extinguishing powder (FEP) compositions based on monoammonium phosphate (MAP). The specific efficiency improvement of fire-extinguishing powders can be achieved through the use of crystalline hydrates containing chemically bound water. Through the process of thermal decomposition of crystalline hydrates, a large amount of water vapor is released, decreasing the concentration of combustible gases and, as a result, reducing the temperature in the combustion zone.

In the present investigation, we propose to use magnesium ammonium phosphate (struvite) hexahydrate MgNH_4_PO_4_·6H_2_O as an extinguishing agent, which is easily accessible, environmentally friendly and simple to synthesize [19,20]. The thermal decomposition of struvite is accompanied by a significant endothermic effect, while the surface of the fire-extinguishing component can act as an adsorbent that temporarily removes the active centers of the flame from the reaction zone [21,22]. To reduce hygroscopicity and caking and to improve the flowability of struvite, it is proposed to use nanosized silicon dioxide as a protective coating on the surface of the crystalline hydrate, imparting hydrophobic properties to the powder composition [23,24,25,26]. In order to obtain an effective functional additive to fire-extinguishing powder compositions based on crystalline hydrates (e.g., struvite), the purpose of this work was to investigate the processes of synthesis and hydrophobization of the silicon dioxide surface with different sizes of monodisperse spherical particles.

## 2. Materials and Methods

The following reagents were used to prepare hydrophobic silicon dioxide samples: tetraethoxysilane (TEOS, (Si(OC_2_H_5_)_4_, Alfa Aesar, Haverhill, MA, USA) as a source of silicon dioxide; polymethylhydrosiloxane (PMHS, (CH_3_(H)SiO)n, 99%, MM = 2500, Alfa Aesar) as a hydrophobizing compound; distilled water, ethyl alcohol (rectified), and ammonium hydroxide (NH_4_OH, 25%, analytical grade, Vekton JSC, Saint Petersburg, Russia) as a reaction medium.

The following precursors were used for struvite synthesis: magnesium chloride, 6-hydrate (MgCl_2_·6H_2_O, chemically pure, Vekton JSC), ammonium hydrophosphate ((NH_4_)_2_HPO_4_, Vekton JSC) and ammonium hydroxide (NH_4_OH, 25%, analytical grade, Vekton JSC).

In order to compare the thermal behavior of struvite with the most common fire-extinguishing powder agent, we used monoammonium phosphate powder (MAP, NH_4_H_2_PO_4_, Vekton JSC) with an average particle size (D50) of ~20 µm, which was obtained by grinding it in a ball mill.

Samples of hydrophobic silica were prepared in two stages, including the synthesis of silicon dioxide with a spherical particle shape, based on the traditional Stöber method, and post-synthetic modification of the surface with a hydrophobic compound (PMHS). 

In order to obtain silicon dioxide samples with different textural properties (particle size and specific surface area), during the first stage, the ratio of water (mol) was varied in respect to other synthesis components:

TEOS 1:H_2_O 32:C_2_H_5_OH 9.5:NH_4_OH 1.4—*S1-0*;

TEOS 1:H_2_O 28:C_2_H_5_OH 9.5:NH_4_OH 1.4—*S2-0*;

TEOS 1:H_2_O 25:C_2_H_5_OH 9.5:NH_4_OH 1.4—*S3-0*;

TEOS 1:H_2_O 22:C_2_H_5_OH 9.5:NH_4_OH 1.4—*S4-0*;

TEOS 1:H_2_O 19:C_2_H_5_OH 9.5:NH_4_OH 1.4—*S5-0*.

The synthesis time was determined from the specific surface area using the example of sample *S1-0* synthesis, where the molar ratio of [TEOS]/[H_2_O] corresponded to the proportion 1/32 and stirring speed of 600 rpm. According to the results, the optimal synthesis time was 45–90 min (Table 1).

During the second stage, postsynthetic grafting of nonpolar groups of a hydrophobizing compound (PMHS) to the surface of the initial silicon dioxide samples *S1-0*–*S5-0* was carried out in hexane at reflux for 4 h.

To obtain silicon dioxide samples with different particle sizes bearing superhydrophobic properties (contact angle > 150°), we experimentally determined the amount of a hydrophobizing agent. Several series of samples were prepared, where the mass fraction of PMHS in relation to the initial silicon dioxide was:

1%—samples *S1-1*, *S2-1*, *S3-1*, *S4-1*, *S5-1*;

3%—samples *S1-3*, *S2-3*, *S3-3*, *S4-3*, *S5-3*;

5%—samples *S1-5*, *S2-5*, *S3-5*, *S4-5*, *S5-5*;

7%—samples *S1-7*, *S2-7*, *S3-7*, *S4-7*, *S5-7*.

The synthesis of the fire-extinguishing component, struvite was carried out according to the following procedure: to a water solution of magnesium chloride (0.1 M) and ammonium hydrogen phosphate (0.1 M), aqueous ammonia was added to pH 9.5. The mixture was stirred with a magnetic stirrer at 25 °C for 5 min. The precipitate was filtered off, washed with distilled water, and dried for a day at 25 °C. 

To perform the rheological properties investigation of the fire-extinguishing powder compositions based on struvite and hydrophobic silicon dioxide, powder mixtures with a mass fraction of 5% of functional additive were prepared. The mixing of composition components was carried out in a ball mill for 3 h to achieve a uniform distribution of functional additive particles on the surface of struvite.

The properties of the obtained samples were studied by various physicochemical methods of analysis. The surface modification of the samples by nonpolar groups was investigated by IR spectroscopy in the region of 400–4000 cm^−1^ on an IFS–66/S IR–Fourier spectrometer (Bruker, Bremen, Germany). Textural properties were determined by low-temperature nitrogen sorption on an ASAP 2020 instrument (Micromeritics, Norcross, GA, USA) after degassing the test sample in a vacuum at 90 °C for 3 h. The specific surface of the samples (*S_BET_*) was determined by the BET method, and the pore size and volume (*V_tot_*) were determined by the BJH method from desorption isotherms. The correlation coefficient in all cases was >0.99.

The amount of reactive silanol groups (mmol/g) on the surface of the initial SiO_2_ samples was determined by thermogravimetric analysis (TGA) in the temperature range of 200–1000 °C using a TGA/DSC 1 instrument (METTLER-TOLEDO, Greifensee, Switzerland) in an air atmosphere at a heating rate of 10 °C/min. Aqueous suspensions of SiO_2_ samples with different particle sizes were preliminarily boiled for 1 h and then dried at 80 °C for 1 h. 

The molar content of OH groups (*n_OH_*) was calculated using the formula:$$n_{OH}=\frac{2(W_{t_{o}}-W_{t})}{100\cdot M_{H_{2}O}}$$
where ($W_{t_{o}}-W_{t}$) is a weight loss (%) in the temperature range 200–1100 °C; $M_{H_{2}O}$ is the molecular weight of water.

The study of the thermal degradation process and the collection and determination of gaseous decomposition products of samples of fire-extinguishing powder compositions were carried out using a TGA/DSC-IST16-GC/MS instrument complex, including a TGA/DSC 3+ (Mettler Toledo, Greifensee, Switzerland), an IST16 sampler (SRA Instruments, Marcy-l’Étoile, France) and an Agilent 7890B/5977B GC/MS spectrometer (Agilent Technologies, Santa Clara, CA, USA). The analysis was carried out in an air atmosphere with a temperature range of 25–600 °C and a heating rate of 10 °C/min.

The analysis of gaseous decomposition products GC/MS was performed using the following parameters: column HP-5ms UI, 30 m, 0.25 µm, 0.25 µm, EI ionization 70 eV, carrier gas helium, flow rate 1 mL/min. Chromatography conditions were as follows: temperature 35 °C for 2 min; evaporator temperature 250 °C; injection volume 250 µL (sampler IST16); flow split 50–100:1. The results of gas composition analysis were processed using MSD ChemStation F01.03.2357 software. For the studied samples, peaks corresponding to ammonia (with *m*/*z* 17) and water (with *m*/*z* 18) were found in ion chromatograms.

The morphology and particle size of the synthesized samples were studied by scanning electron microscopy on an FEI Quanta FEG650 instrument (ThermoFisher Scientific, Breda, The Netherlands).

The contact angle on the surface of preliminarily pressed samples was estimated using a DSA100 laboratory drop shape analyzer (KRÜSS, Hamburg, Germany) with the volume of water droplets equal to 8 μL. The average value of the contact angle was calculated by the formula *θ* = *θn*/*n*, where *n* is the number of measurements (at least 10 for each material sample).

To estimate the particle size distribution of powder compositions, a HELOS/KR laser diffraction analyzer (Sympatec GmbH, Clausthal-Zellerfeld, Germany) with a measurement range from 0.9 µm to 3.5 mm was utilized. This instrument allows dry dispersion of the sample with compressed air (0.5–5 bar).

An FT4 powder rheometer (Freeman Technology, Tewkesbury, UK) was used to evaluate the rheological characteristics of fire-extinguishing powder samples by determining their cohesion and flow-function coefficient utilizing the shear test method [19]. The shear test procedure was carried out as follows: the tested powder composition was compacted and preliminary shears of the powder layer were performed until the constant flow point was reached and until the powder began to flow. The preliminary shearing of the powder was repeated 4–5 times using a normal stress of 20–80% of the compaction stress. Shear test results were interpreted using the Mohr stress circle (Figure 1) [27,28].

Points on the circle correspond to the values of normal (abscissa) and shear (ordinate) stresses of bulk material. According to the results of shear tests at different normal loads on the FEP sample, compacted under the same force, a yield stress line was obtained, which expresses the ratio between the powder shear resistance and the normal compressive load. Extrapolation of this line to the y-axis corresponds to cohesion (*τ_c_*, shear stress at zero normal load). Cohesion reflects the influence of autohesion on the flow of coarse materials, for which the yield strength line does not pass through the origin. The angle between the yield lines and the abscissa corresponds to the internal friction that occurs at the contact points during the mutual movement of powder particles. Then, the highest stress during compaction of the sample of powder material (*σ*_1_) and the stress of free flow of the FEP (*σ_c_*) were determined. The ratio of *σ*_1_ and *σ_c_* makes it possible to determine the flow function coefficient (*ff_c_*), which qualitatively characterizes the powder flow conditions (Table 2) [19].

## 3. Results

The determination of the texture property results of silicon dioxide samples, which were prepared with different amounts of water, are presented in Table 3.

The nitrogen sorption-desorption isotherms in the initial silica samples (Appendix A, Appendix A) correspond to type II (UPAC). These types of isotherms are typical for low-porous materials. The horizontal part of the sorption isotherms in the range of relative pressures of 0–0.8 for the initial silicon dioxide samples indicates the presence of micropores. The microporosity of samples is also confirmed by the pore size distribution, where the distribution curves correspond to the microporous region (Appendix A, Appendix A). Moreover, the proportion of micropores increases with increasing particle size (Table 3). It should be noted that sample S1-0, which was synthesized with the maximum amount of water, has the largest pore volume. In this case the interparticle distances are probably comparable with the diameter of SiO_2_ particles (Appendix A, Appendix A, *S1-0*). 

The influence of the water amount used in SiO_2_ synthesis by the Stöber method on particle size is shown in Figure 2. The Stöber method is known to include two successive stages: first, the hydrolysis of TEOS, accompanied by condensation of silanol monomers, leading to the formation of silicon dioxide particles (1); and second, the subsequent growth of small particles of silicon dioxide due to the deposition of silanol monomers on their surface (2) [20]. Therefore, the water amount increase during the synthesis of silicon dioxide promotes the hydrolysis of TEOS, preventing the growth of particles. Conversely, a decrease in the water amount reduces the TEOS hydrolysis rate, leading the primary SiO_2_ particles to “overgrow” with silanol monomers, increasing in size through continuous ionic interaction. Since the same ratio of TEOS to ammonia (a catalyst for the hydrolysis reaction) was maintained during the synthesis of all samples, the [H_2_O]/[C_2_H_5_OH] ratio seems to be the determining factor in the rate of hydrolysis and particle growth.

According to Figure 3, hydrophobization of silicon dioxide modified with 5% PMHS does not lead to a significant change in particle size. At the same time, the values of the main textural characteristics (specific surface area and total pore volume) of the studied samples after their surface hydrophobization predictably decrease (Table 4, Appendix A, Appendix A). Appendix A demonstrates that the sorption isotherms and pore size distribution curves in hydrophobized samples (*S1-7*–*S5-7*) repeat the shape of the sorption isotherms of the initial silicon dioxide samples.

Figure 3 presents the IR spectra of hydrophobic silicon dioxide with different particle sizes for samples with minimum (1%) and maximum (7%) PMHS amounts. Figure 3 reveals that in almost all cases, 1% PMHS is not enough to hydrophobize the surface of silica samples, which was also confirmed by the contact angle determination results (Table 5). The samples with small particles *S1-1* and *S2-1* are exceptions, for their IR spectra contain absorption bands of low intensity at 2914 cm^−1^, which, most likely, corresponds to the stretching vibrations of CH groups. According to the results of contact angle determination, these samples are hydrophobic even with a small amount of PMHS (1%) (Table 5).

The IR spectra of the samples with 7% of a hydrophobizing agent demonstrate the presence of bonds confirming the modification of silicon dioxide surface samples by nonpolar PMHS fragments. The new formation of bonds through silanol groups is confirmed by the intensity reduction of the ~3440 cm^−1^ band, which is associated with the superposition of the absorption of the stretching vibrations with the Si–OH group, as well as the 960 cm^−1^ band shift to the low-frequency region (948–931 cm^−1^), which characterizes Si–O vibrations in pure silicon dioxide. The absorption bands at 899–908 cm^−1^ and 836–837 cm^−1^ can be referred to Si–C bond vibrations. The absorption bands in the range 2852–2981 cm^−1^ correspond to symmetric and asymmetric stretching vibrations of the C–H alkyl bond in the SiCH_3_, SiCH_2_, and SiCH groups. The absorption band 2169–2171 cm^−1^ corresponds to the Si–H bond in the O–SiH_3_, O_2_–SiH_2_ and O_3_–SiH groups.

Data given in Table 5 confirm that 1% of PMHS for hydrophobization of the samples surface with large particle sizes *S4-0* and *S5-0* is insufficient—the samples remain hydrophilic. This might be associated with a large number of silanol groups which remain free after modification (determined by TGA, Table 1). Based on contact angle determination results, we experimentally established that the required PMHS amount increase for more than 5% does not significantly affect hydrophobization. The amount of 5% PMHS in almost all cases is sufficient to achieve the hydrophobic behavior of the surface. At the same time, samples *S1-5*, *S4-5*, *S4-7*, *S5-3*, *S5-5* and *S5-7*, with the contact angle more than 150°, can be classified as superhydrophobic.

Powder compositions based on struvite and functional additive samples with different particle sizes were prepared—*S1-5* (FEP-1), *S3-5* (FEP-2), *S5-5* (FEP-3) for thermal and rheological studies.

In the process of the current research investigation, we found that the uniform coating of a struvite surface with a functional additive is a crucial factor affecting the flowability of fire-extinguishing powders (Table 6, Figure 4 and Figure 5). Table 6 data show that FEP-1 with *S1-5* particle size ~50 nm (as a functional additive) is classified as free-flowing (cohesion 0.42 kPa, *ff_c_* > 10) compared to pure struvite (cohesion 2.53 kPa, *ff_c_* < 2).

By comparing SEM images of fire-extinguishing powders FEP1 (*S1-5*) and FEP2 (*S3-5*), it becomes clear that as the functional filler particle diameter increases, the coating on the struvite surface becomes less uniform, which leads to the increase and cohesion growth of the number of contacts between particles.

The protective coating formation of superhydrophobic silicon dioxide particles on the surface of struvite significantly improves the water-repellent properties of the crystalline hydrate compared to the unmodified compound. At the same time, the particles surface of powder composition samples is characterized by a contact angle of more than 140° (Figure 6, Table 7).

Figure 7 reveals that the sizes of struvite particle aggregates, without the use of a functional additive, are up to 850 µm (D(50) = 291 µm, D(90) = 661 µm, D(99) = 859 µm). When hydrophobic silicon dioxide *S1-5* (5% wt.) is introduced into the powder composition, the size of the aggregates of fire-extinguishing composition particles does not exceed 15 microns. The size parameter of particle aggregates D50 in this case was 3.63 µm; D90 = 9.00 µm; D99 = 14.88 µm.

Compositions of fire-extinguishing powder based on struvite (FEP-1) and MAP (FEP-0) with a silicon dioxide particle size of 50 nm as a functional additive were used for thermal analysis. 

The results of thermal analysis (Figure 8) revealed that the process of thermal destruction of struvite proceeds in one stage according to the following scheme:2MgNH_4_PO_4_ 6H_2_O → Mg_2_P_2_O_7_ + 7H_2_O↑ + 2NH_3_↑

Thermal decomposition of MAP is a multi-stage process and starts at a higher temperature:$$2{\mathrm{NH}}_{4}H_{2}{\mathrm{PO}}_{4}\overset{140{}^{\circ}C}{\to }{\left({\mathrm{NH}}_{4}\right)}_{2}H_{2}P_{2}O_{7}+H_{2}O$$
$$3{\mathrm{NH}}_{4}H_{2}{\mathrm{PO}}_{4}\overset{170{}^{\circ}C}{\to }{\left({\mathrm{NH}}_{4}\right)}_{3}H_{2}P_{3}O_{10}+2H_{2}O$$
$${\mathrm{NH}}_{4}H_{2}{\mathrm{PO}}_{4}\overset{190{}^{\circ}C}{\to }{\mathrm{NH}}_{4}{\mathrm{PO}}_{3}+H_{2}O$$
$${\mathrm{NH}}_{4}H_{2}{\mathrm{PO}}_{4}\overset{250{}^{\circ}C}{\to }{\mathrm{HPO}}_{3}+{\mathrm{NH}}_{3}+H_{2}O$$

The end effects for fire-extinguishing powder compositions FEP-0 and FEP-1 were determined according to the DSC curves (Figure 8). Figure 8B shows that the endothermic effect of struvite destruction in the temperature range ~50–194 °C is accompanied by heat absorption of ~1400 J/g. In comparison, the endothermic effect of MAP decomposition (Figure 8A) is significantly lower (~800 J/g) than for struvite (Table 8).

The gaseous products of the decomposition of fire-extinguishing powders were taken for analysis. According to this calculation, the ratios of released gases are presented in Table 9.

The analysis results of the thermal decomposition of the gaseous products of fire-extinguishing powders indicate a high proportion of water (~97%) released during the decomposition of a struvite based composition compared to the sample based on MAP (~90%).

## 4. Conclusions

Conditions for the synthesis and hydrophobization of a silicon dioxide surface with different particle sizes of spherical shape were studied in this work for their promising application as a functional additive in the preparation of fire-extinguishing powders based on crystalline hydrates. The [H_2_O]/[C_2_H_5_OH] ratio is determined to be a crucial factor for the regulation of the size of silicon dioxide particles during their synthesis by the Stöber method.

Textural and structural properties of silicon dioxide samples with different particle sizes were studied by the method of low-temperature sorption. The process of hydrophobization does not significantly affect the size and shape of the particles. The textural characteristics of hydrophobized silica samples (specific surface area and total pore volume) depend on the amount of the hydrophobizing compound PMHS. 

We experimentally established that 5% of the hydrophobizing compound (PMHS) allows for sufficient modification of the surface of silicon dioxide samples, regardless of the particle size. The TGA analysis determined that silicon dioxide samples with a large particle size (300–400 nm) have a higher content of reactive silanol groups on the surface of SiO_2_. The same samples modified with 5–7% PMGS, according to the contact angle determination results, are superhydrophobic.

We demonstrated that the utilization of hydrophobic silicon dioxide with a spherical particle size of ~50 nm as a functional additive prevents struvite fromcaking, reduces the size of its agglomerates, and produces free-flowing properties. It was confirmed that the particle size of the functional additive and their uniformity on the crystalline hydrate surface is a crucial factor for the preparation of struvite based fire-extinguishing compositions. The uniform distribution of the functional additive on the surface of the struvite forms a protective coating that guarantees the superhydrophobic state of the crystalline hydrate with a contact angle of more than 150°.

Thermal behavior research on struvite-based powder compositions and hydrophobic silicon dioxide confirms the promising application of crystalline hydrates as components of fire-extinguishing powders. The amount of heat absorbed during the thermal destruction of struvite is at least two times higher than during the decomposition of monoammonium phosphate, which is currently the main component of fire-extinguishing powder compositions.

## Figures and Tables

**Figure 1 nanomaterials-13-01186-f001:**
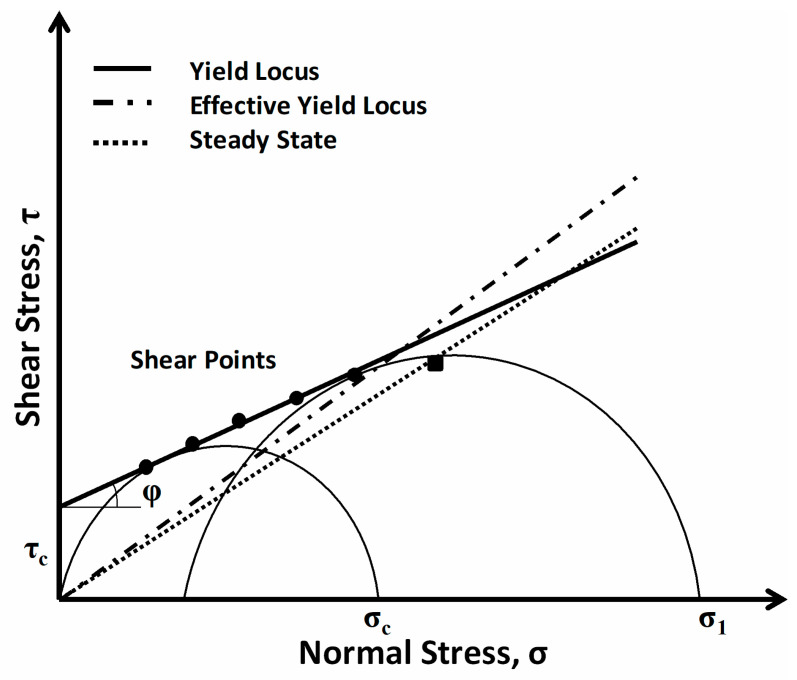
Results of the shear test using the Mohr stress circle.

**Figure 2 nanomaterials-13-01186-f002:**
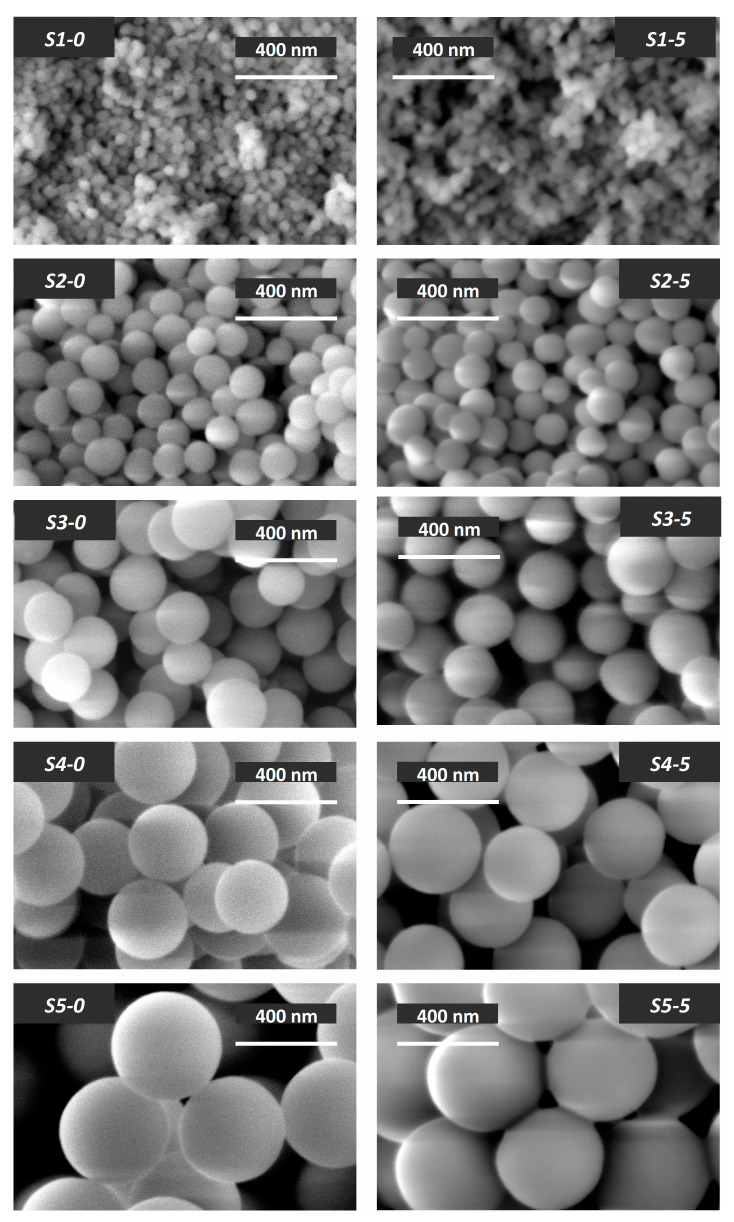
SEM images of silicon dioxide initial samples (**left**) and the same samples hydrophobized with 5% PMHS (**right**).

**Figure 3 nanomaterials-13-01186-f003:**
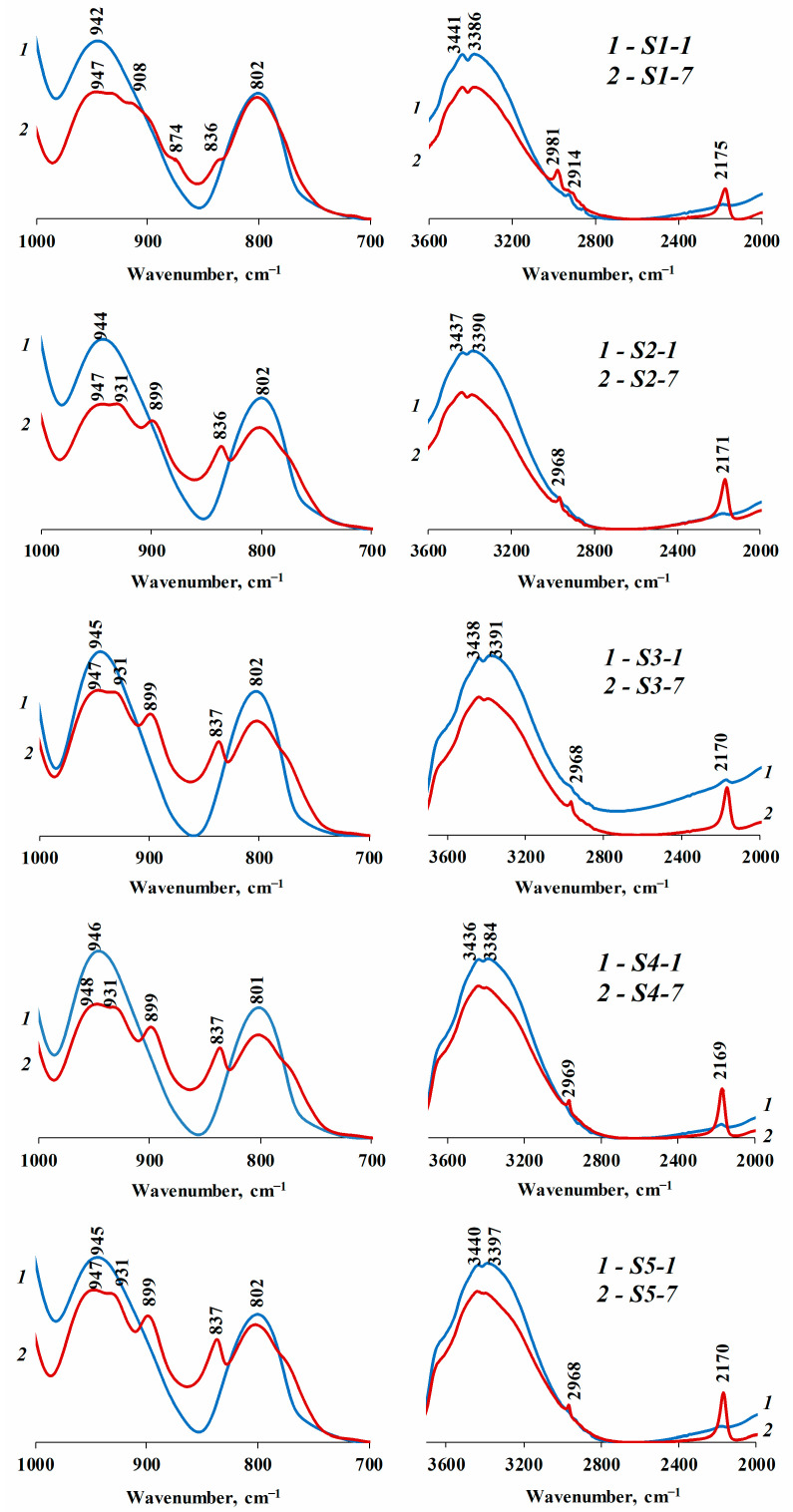
IR spectra of silicon dioxide samples with different particle sizes, hydrophobized with 1% (1) and 7% (2) of PMHS.

**Figure 4 nanomaterials-13-01186-f004:**
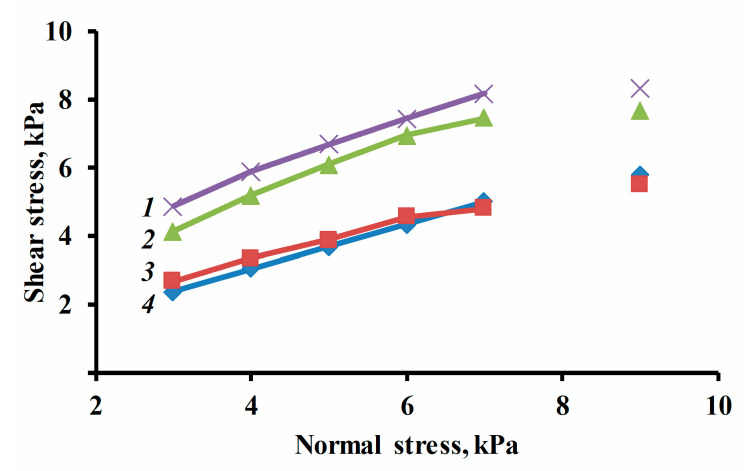
The shear test results of the composition of samples of fire-extinguishing powder: 1—struvite, 2—FEP-1, 3—FEP-2, 4—FEP-3.

**Figure 5 nanomaterials-13-01186-f005:**
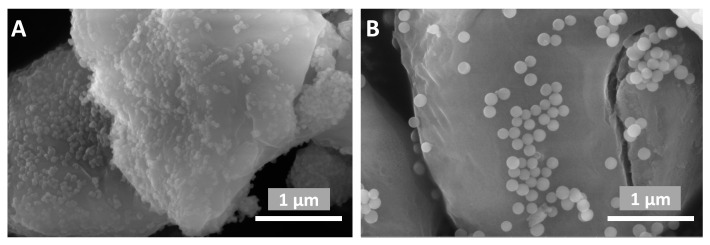
SEM images of fire-extinguishing powder compositions based on struvite and functional additives (**A**) FEP-1 and (**B**) FEP-2.

**Figure 6 nanomaterials-13-01186-f006:**
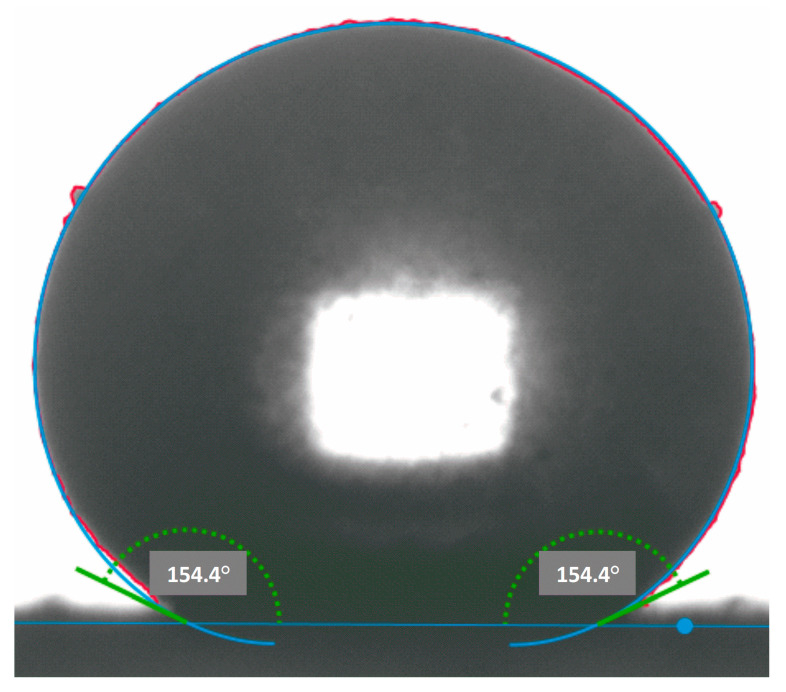
Water contact t angle determination on the surface of fire-extinguishing powder FEP-3 sample.

**Figure 7 nanomaterials-13-01186-f007:**
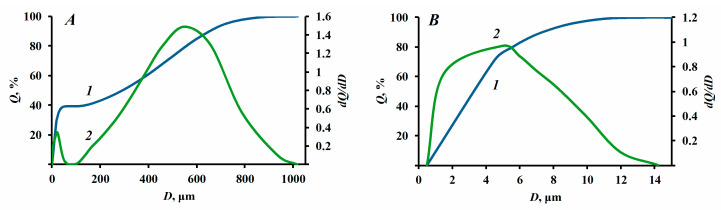
Grain size distribution of struvite samples (**A**); struvite based fire-extinguishing powder composition (**B**); where 1—integral distribution, 2—differential distribution.

**Figure 8 nanomaterials-13-01186-f008:**
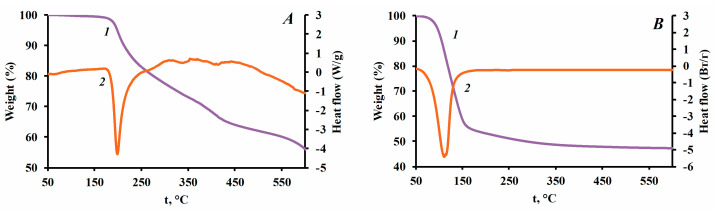
Results of thermal tests on samples of fire-extinguishing powders (**A**) FEP-0; FEP-1 (**B**); where 1—TGA, 2—DSC.

**Table 1 nanomaterials-13-01186-t001:** Textural properties of sample S1-0 as a function of synthesis time.

No	Synthesis Time, min	*S_BET_*, m^2^/g	*S_micropores_* *(t-Plot)*, m^2^/g	*V_tot_*, cm^3^/g	*V_micropores_* *(t-Plot)*, cm^3^/g	Pore Diameter (BJH), nm	
						Adsorption	Desorption
1	15	218 ± 3.7	23	0.56	0.016	10	9
2	45	238 ± 4.7	40	0.88	0.027	11	16
3	60	255 ± 5.6	44	0.97	0.031	14	16
4	90	202 ± 3.9	24	0.82	0.018	17	16
5	120	178 ± 2.6	7	0.69	0.006	13	12

**Table 2 nanomaterials-13-01186-t002:** Classification of powder materials according to the flow function coefficient.

Flow Function Coefficient (*ff_c_*)	Powder Flow Characteristics
*ff_c_* < 1	Absence of flow
*1 < ff_c_* < 2	Bad flow
*2 < ff_c_* < 4	Obstructed flow
*4 < ff_c_* < 10	Easy flowing
*ff_c_* >10	Free flowing

**Table 3 nanomaterials-13-01186-t003:** Textural characteristics of initial silicon dioxide samples.

Sample	H_2_O/TEOS	*S_BET_*, m^2^/g	*S_micropores_* *(t-Plot)*, m^2^/g	*V_tot_*, cm^3^/g	*D_por_*, nm	*D*, nm	Mass Loss 200–1000 °C, % (TGA)	Amount of Silanol Groups, mmol/g
*S1-0*	32	255 ± 5.6	44	0.97	14	50	3.7	4.11
*S2-0*	28	225 ± 7.6	134	0.37	20	120	3.9	4.33
*S3-0*	25	232 ± 8.4	151	0.26	18	200	4.1	4.56
*S4-0*	22	249 ± 9.4	190	0.17	6	300	4.3	4.78
*S5-0*	19	34 ± 0.26	27	0.05	8	400	4.5	5.00

**Table 4 nanomaterials-13-01186-t004:** Textural characteristics of hydrophobized silica samples.

Sample	H_2_O/TEOS Ratio	Amount of PMHS, %	*S_BET_*, m^2^/g	*V_tot_*, cm^3^/g	*D_pore_*, nm
*S1-1*	32	1	74.3 ± 1.3	0.33	18
*S2-1*	28		28.9 ± 0.5	0.24	33
*S3-1*	25		22.2 ± 0.1	0.14	25
*S4-1*	22		10.6 ± 0.2	0.04	16
*S5-1*	19		7.6 ± 0.2	0.02	12
*S1-3*	32	3	54.3 ± 0.5	0.3	22
*S2-3*	28		23.3 ± 0.4	0.23	39
*S3-3*	25		15.9 ± 0.3	0.09	22
*S4-3*	22		8.3 ± 0.2	0.03	14
*S5-3*	19		6.7 ± 0.2	0.02	8
*S1-5*	32	5	52.9 ± 0.7	0.29	22
*S2-5*	28		15.9 ± 0.3	0.19	47
*S3-5*	25		21.8 ± 0.3	0.12	23
*S4-5*	22		13.1 ± 0.3	0.02	7
*S5-5*	19		6.0 ± 0.2	0.01	7
*S1-7*	32	7	47.3 ± 0.5	0.27	23
*S2-7*	28		19.3 ± 0.4	0.20	41
*S3-7*	25		5.3 ± 0.2	0.04	26
*S4-7*	22		4.5 ± 0.1	0.02	19
*S5-7*	19		3.7 ± 0.2	0.01	6

**Table 5 nanomaterials-13-01186-t005:** Contact angle for silicon dioxide samples with different particle sizes depending on the amount of PMHS.

No	*D*, nm	Amount of PMHS							
		1%		3%		5%		7%	
		No	*θ*, °	No	*θ*, °	No	*θ*, °	No	*θ*, °
*S1-0*	50	*S1-1*	143.4 ± 1.2	*S1-3*	147.3 ± 1.1	*S1-5*	152.6 ± 1.3	*S1-7*	142.0 ± 1.6
*S2-0*	120	*S2-2*	143.3 ± 1.6	*S2-3*	144.9 ± 1.8	*S2-5*	146.6 ± 1.5	*S2-7*	142.5 ± 1.6
*S3-0*	200	*S3-3*	130.4 ± 1.2	*S3-3*	135.8 ± 1.5	*S3-5*	144.6 ± 1.3	*S3-7*	146.4 ± 1.5
*S4-0*	300	*S4-4*	- *	*S4-3*	147.2 ± 1.6	*S4-5*	150.9 ± 1.1	*S4-7*	152.8 ± 0.8
*S5-0*	400	*S5-5*	-	*S5-3*	158.9 ± 1.7	*S5-5*	160.9 ± 1.6	*S5-7*	157.4 ± 1.5

* Material absorbs water instantly.

**Table 6 nanomaterials-13-01186-t006:** Rheological characteristics of fire-extinguishing powder compositions.

FEP Sample	Functional Additive	Cohesion, kPa	Flow Function Coefficient (*ff_c_*)
struvite	-	2.53 ± 0.05	1.75
FEP-1	*S1-5*	0.42 ± 0.03	10.70
FEP-2	*S3-5*	1.18 ± 0.01	3.87
FEP-3	*S5-5*	1.72 ± 0.02	2.37

**Table 7 nanomaterials-13-01186-t007:** Water contact angle on the surface of composition samples of fire-extinguishing powder.

Sample	*θ,* °
FEP-1	150.5 ± 2.7
FEP-2	140.4 ± 1.4
FEP-3	155.2 ± 1.8

**Table 8 nanomaterials-13-01186-t008:** Thermal tests of samples of fire-extinguishing powders.

Sample	Total Weight Loss, %	Initial Effect Temperature, °C	Final Effect Temperature, °C	Thermal Effect, J/g
FEP-0	63.0	154.4	308.2	−773.5
FEP-1	53.1	58.0	200.1	−1432.7

**Table 9 nanomaterials-13-01186-t009:** The gaseous products ratio of fire-extinguishing powders samples thermal decomposition.

The Gaseous Products Ratio	FEP-0	FEP-1
NH_3_, %	10.6	3.5
H_2_O, %	89.4	96.5

## Data Availability

Data will be made available on request.

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
