# Peer review of "Synthesis of Hydrophobic Nanosized Silicon Dioxide with a Spherical Particle Shape and Its Application in Fire-Extinguishing Powder Compositions Based on Struvite"

_nanomaterials, 2023, doi:10.3390/nano13071186_

Round 1

Reviewer 1 Report

The manuscript by I. V. Valtsifer et al. discusses the surface modification of struvite with hydrophobized silica particles generated by the Stoeber process. Struvite is a promising fire-extinguishing material, but its powder is not free-flowing and is prone to agglomeration. The Stoeber process is well known and established, and a part of the present work reiterates what is known for a long time, i.e. that you can get larger particles when you have more water in the process. The hydrophobic coating of the SiO2 particles with a siloxane (PMHS) and their use to coat struvite surfaces is the main innovation of the paper. The successful coating of the SiO2 particles with PMHS is properly demonstrated. The Mohr stress cycle is used to examine the flow properties of the struvite powder and it is convincingly demonstrated that the flow function coefficient of the material increases when it is coated with hydrophobized SiO2. The coated material appears to behave very well in thermal tests.

Overall, I find the work useful and innovative, and the manuscript well written, with an effective use of a large number of techniques. I support its publication. Some minor revisions are in order as follows:

-          It would be useful to insert error bars wherever possible. Error bars can only be found in Table 5, but not in the other tables. I would expect to see error bars in the BET areas and pore volumes and diameters.

-          The equation used to calculate the amount of silanol groups is based on the assumption that a water molecule is subtracted from two (adjacent) silanol groups. The quantitation of silanol groups is of course possible with other methods (NMR, IR etc) and I wonder if the authors checked their numbers with alternative methods.

-          In Figures 2 and 4 we see pore size distributions for the various SiO2 materials. In some cases, one does not see any clear size distribution or any well-defined average size, yet the tables contain average diameters (without error bars). What is the meaning of these diameters?

-          Line 252. Replace Table 3 with Table 5

-          Figure 5 abscissa. Change Wavenumbed to Wavenumber!

Author Response

Dear Reviewer,

Thank you for the time and effort that you dedicated to providing feedback on our manuscript “Synthesis of Hydrophobic Nanosized Silicon Dioxide with a Spherical Particle Shape and Its Application in Fire-extinguishing Powder Compositions Based on Struvite” for publication in MDPI Nanomaterials. Our author group is grateful for the insightful comments on and valuable improvements to our paper. We have incorporated most of the suggestions made by the reviewers. Those changes are highlighted within the manuscript. Please see below for a point-by-point response to the comments and concerns. All page numbers refer to the revised manuscript file with tracked changes.

Questions:

  1. It would be useful to insert error bars wherever possible. Error bars can only be found in Table 5, but not in the other tables. I would expect to see error bars in the BET areas and pore volumes and diameters.

Answer:

Tables 1 (page 3), 3 (page 6) and 4 (page 9) show the error bars in determining the specific surface area by the BET method. The correlation coefficient in all cases was >0.99 (page 4).

  1. The equation used to calculate the amount of silanol groups is based on the assumption that a water molecule is subtracted from two (adjacent) silanol groups. The quantitation of silanol groups is of course possible with other methods (NMR, IR etc) and I wonder if the authors checked their numbers with alternative methods.

Answer:

The number of silanol groups was estimated only by thermogravimetry method according to the formula given in the work. Other methods, in this case, were not used.

  1. In Figures 2 and 4 we see pore size distributions for the various SiO2 In some cases, one does not see any clear size distribution or any well-defined average size, yet the tables contain average diameters (without error bars). What is the meaning of these diameters?

Answer:

Figures 2 and 4 show the pore distribution data, the sizes of which were determined using the ASAP 2020 instrument (Micromeritics, USA). In this case, it is difficult to insert error bars in the determination of the pore size, since there are not many powder materials with a narrow pore size distribution (zeolites, mesoporous silica materials such as MCM, SBA, etc., some carbon materials). The bulk of powder materials has a wide pore size distribution.

In our case, the article emphasizes that spherical silica particles are microporous, what is confirmed by the t-Plot method (Table 1). For better understanding, a column with micropore area values has been added to Table 3 (page 6). Moreover, the proportion of micropores increases with increasing particle size (Table 3). This trend can also be seen in Figure 2 - a section of the curve near the dV/dD axis.

The average pore values determined by the BJH method, as it is assumed in the text of the article, includes the distance between the particles.

Perhaps these values are indeed uninformative, so they can be excluded from tables. And the distribution curves, in our opinion, should be left.

  1. Line 252. Replace Table 3 with Table 5

Answer:

The alteration has been made to the article text (page 10).

  1. Figure 5 abscissa. Change Wavenumbed to Wavenumber!

Answer:

Figure 5 has been corrected (page 11).

Best regards,

Yan Huo (Corresponded Author)

College of Aerospace and Civil Engineering, Harbin Engineering University,
Harbin 150001, Heilongjiang, China

E-mail: huoyan205@hotmail.com

Reviewer 2 Report

The paper  reports about the obtainement of an interesting application supported by a large numeber of accurate  experimental data. Even if not exceptionally innovative it deserves publication after some minor revision.Indeed the paper reports about series of systematic  results  which could be described more synthetically and some part moved  to the additional information to make easier the reading and better focus on the real new results.

Some suggestiions

1.Figure 1 could be removed is   well known

2. Emphasis shoiìul be made along the text about the nain objective of improving  flame retardancy performances where  the preparation of hydrophobic silica particles is an useful but secondary aspect.

3.The replacement of fig 2 annd fig 4 with two synthwtic table would facilitate the reader.The same couls also be valid for fig 5

4.The last sense of the abstract is vague and not related to the work done.More information should be added on flammability tests.(see conclusion)

4.

Author Response

Dear Reviewer,

Thank you for the time and effort that you dedicated to providing feedback on our manuscript “Synthesis of Hydrophobic Nanosized Silicon Dioxide with a Spherical Particle Shape and Its Application in Fire-extinguishing Powder Compositions Based on Struvite” for publication in MDPI Nanomaterials. Our author group is grateful for the insightful comments on and valuable improvements to our paper. We have incorporated most of the suggestions made by the reviewers. Those changes are highlighted within the manuscript. Please see below for a point-by-point response to the comments and concerns. All page numbers refer to the revised manuscript file with tracked changes.

Questions:

  1. Figure 1 could be removed is well known.

Answer:

From our point of view, for a better understanding of the data in Table 2, where the classification of powder materials is given according to the flow function coefficient, Figure 1 should be left.

  1. Emphasis should be made along the text about the main objective of improving flame retardancy performances where the preparation of hydrophobic silica particles is a useful but secondary aspect.

Answer:

Figures 2 and 4, in accordance with your recommendations, have been removed from the main text of the article and placed in the section of Supporting Information.

  1. The replacement of fig 2 and fig 4 with two synthetic tables would facilitate the reader. The same could also be valid for fig 5.

Answer:

Figure 5 (in the corrected version - Figure 3), from our point of view, is a clearer presentation of the surface hydrophobization of the SiO2 particles than the data placed in the table.

  1. The last sentence of the abstract is vague and not related to the work done. More information should be added on flammability tests (see conclusion).

Answer:

The abstract content has been corrected (page 1).

Best regards,

Yan Huo (Corresponded Author)

College of Aerospace and Civil Engineering, Harbin Engineering University,
Harbin 150001, Heilongjiang, China

E-mail: huoyan205@hotmail.com

Reviewer 3 Report

This paper investigates the textural and morphological characteristics of hydrophobic silicon dioxide particles obtained by hydrolyzing TEOS and modifying it with hydrophobic polymethylhydrosiloxane. The experiments have been fully performed, and valid data have been obtained. Please add the following.

1) Specify the volume of water droplets in the contact angle measurement.

2) Specify the number n, the number of contact angle measurements.

3) Line 281: Why is increasing the amount of PMHS added by more than 5% have no significant effect on hydrophobicity? Please comment.

Author Response

Dear Reviewer,

Thank you for the time and effort that you dedicated to providing feedback on our manuscript “Synthesis of Hydrophobic Nanosized Silicon Dioxide with a Spherical Particle Shape and Its Application in Fire-extinguishing Powder Compositions Based on Struvite” for publication in MDPI Nanomaterials. Our author group is grateful for the insightful comments on and valuable improvements to our paper. We have incorporated most of the suggestions made by the reviewers. Those changes are highlighted within the manuscript. Please see below for a point-by-point response to the comments and concerns. All page numbers refer to the revised manuscript file with tracked changes.

Questions:

  1. Specify the volume of water droplets in the contact angle measurement.

Answer:

The volume of water droplets in the contact angle measurement was 8 μl (page 4).

  1. Specify the number n, the number of contact angle measurements.

Answer:

The number of contact angle measurements was at least 10 for each material sample (page 4).

  1. Line 281: Why is increasing the amount of PMHS added by more than 5% have no significant effect on hydrophobicity? Please comment.

Answer:

Experiments have shown that the PMHS concentration of 5% is the optimum concentration to provide full surface coverage of silica samples, regardless of their textural characteristics, with a water-repellent layer, and this corresponds to maximum hydrophobicity.

Best regards,

Yan Huo (Corresponded Author)

College of Aerospace and Civil Engineering, Harbin Engineering University,
Harbin 150001, Heilongjiang, China

E-mail: huoyan205@hotmail.com
